# Mobile Apps for Helping Informal Caregivers: A Systematic Review

**DOI:** 10.3390/ijerph18041702

**Published:** 2021-02-10

**Authors:** Marina Sala-González, Virtudes Pérez-Jover, Mercedes Guilabert, José Joaquín Mira

**Affiliations:** Health Psychology Department, Miguel Hernández University, Altamira Building, Avda de la Universidad s/n. 03202 Elche, Spain; marinasala176@hotmail.com (M.S.-G.); mguilabert@umh.es (M.G.); jose.mira@umh.es (J.J.M.)

**Keywords:** caregivers, informal caregivers, mobile applications, telemedicine, mHealth

## Abstract

(1) Background: The physical and psychological consequences suffered by informal caregivers have been extensively studied. MHealth solutions appear to be an opportunity to help overcome the caregiver burden. The objective of this study was to evaluate available mobile applications for informal caregivers of people who are ill and to determine whether these mobile applications were developed considering the needs of caregiver users. (2) Methods: A systematic review was carried out using the MEDLINE, ProQuest, and Scopus databases. The information about mobile applications for informal caregivers was analyzed. This review examined studies published between January 2011 and July 2020 in English. The data extracted from each paper included the development of the mobile application, if that application was assessed considering the caregivers’ needs, functions of the mobile application, measures for evaluating caregivers’ needs, measures for evaluating the effectiveness of the mobile application, and the main results obtained. (3) Results: Eleven studies fulfilled the inclusion criteria. The most common functions of the apps were summaries with information about the person they care for, educational information, resources and services for caregivers, solutions to common problems during care, and questionnaires to assess caregivers’ well-being. Most of these studies assessed caregivers’ needs before designing mobile applications to adapt them to the needs of their users. (4) Conclusions: Mobile applications for caregivers appear to provide solutions for them. Moreover, the effectiveness of these apps will depend largely on whether their characteristics match users’ needs. Current studies have shown the poor quality of evidence.

## 1. Introduction

The increase in life expectancy due to advances in healthcare leads to an increase in chronic diseases and dependent people. This results in the need for support from others to manage the disease and treatment and, in the most severe cases, to carry out the activities of daily living [1,2]. Most of the time it is a family member who takes responsibility for the care. These caregivers are known as informal caregivers as they are not paid for the assistance they provide and often do not have the skills or knowledge to provide care [3,4,5].

In Spain, the most common informal carer is a woman, with an average age of over 50, married and she is usually a daughter or spouse of the person in need of care [6,7]. This caregiver profile is also common in other countries [8].

Informal caregivers take on a variety of caregiving tasks such as addressing physical and emotional needs, managing medications and managing medical appointments [4]. Previous studies have shown the stressful experience of caring for a family member, as it has serious consequences for physical and mental health [9,10]. The main problems faced by informal caregivers are somatic symptoms, depression, anxiety, loneliness, and stress, as well as work and socio-economic problems that give rise to a poorer quality of life [3,4,10].

These caregiver needs have been widely studied. The most cited were carers’ health, especially mental health, time demands, difficulties in handling multiple medications and their side effects, having to coordinate different health professionals, dealing with the emotional burden, lack of leisure time, impact on social relationships, lack of knowledge and information about the illness and treatment, and difficulties in accessing resources to support caregivers [11,12,13].

To reduce the negative consequences of care and increase carers’ quality of life, some interventions have been implemented that have proven to be effective. These measures include: psychoeducational, psychotherapeutic, self-help or multi-component interventions providing education about the disease, problem-solving practices, communication skills, social support or mindfulness [1,10,14,15]. However, these interventions are often expensive, they are not accessible to everyone, and caregivers do not have time for them [9,16].

With the increase of smartphone ownership and their use for health care, there has been an increase in the development of mobile health applications (mHealth) [17]. It is estimated that the number of mobile health applications exceeds 259,000 [1,17]. A new kind of mobile applications is aimed at informal caregivers. These mobile applications can help with providing educational information, remembering doctors’ appointments, coordinating care among all caregivers, managing medication, among other features. To increase their effectiveness, it is also necessary to take into account the needs of users during their development. In this way, these mobile applications constitute a potential resource that offers their users the necessary skills to carry out the tasks of care [9,18]. In addition, mobile applications are shown to be an effective method for monitoring seniors with multiple chronic conditions [16].

Satisfying caregivers’ needs may reduce the burden on the carer [5]. For this reason, it is considered necessary to adapt these mobile applications to the needs of caregivers, taking into account their preferences when developing mobile applications for them.

Therefore, the objectives of this study were to evaluate available mobile applications for informal caregivers of people who are ill and to determine whether these mobile applications were developed considering the needs of caregiver users.

## 2. Methods

### 2.1. Search Strategy

A systematic review was carried out following the Preferred Reporting Items for Systematic Reviews and Meta-Analysis declaration guideline [19]. We search in the MEDLINE, ProQuest, and Scopus databases using MeSH or keywords associated with mobile applications and caregivers to identify all relevant studies and using the Boolean indicators OR and AND (“telemedicine” [MeSH Terms]) OR telemedicine [Title/Abstract] OR “mobile applications” [MeSH Terms] OR mobile applications [Title/Abstract] OR “smartphone” [MeSH Terms] OR smartphone [Title/Abstract]) AND (“caregivers” [MeSH Terms] OR caregivers [Title/Abstract]). The search for documents was limited to publications from January 2011 through July 2020. Moreover, the reference list of selected articles was explored further to find any additional appropriate articles.

### 2.2. Inclusion and Exclusion Criteria

The inclusion criteria for this review were: research published in English that provided results about the design and the evaluation of mobile applications for informal caregivers regardless of the pathology of the patient who needs care and of users’ age. It included both quantitative and qualitative studies, as well as articles with descriptive and experimental methodologies.

We excluded studies that did not create the mobile application, studies about mobile applications for both patients and caregivers, for professionals and for caregivers of people who do not have an illness (e.g., improving young children’s nutrition). Furthermore, we excluded articles where the method was to search for available mobile applications for caregivers.

### 2.3. Data Extraction and Quality Appraisal

The information from each paper was extracted and entered into an Excel program including author data, year, country, objective, participants, target groups, design of the study, duration, development of the mobile application, and if the mobile application was assessed taking into account the caregivers’ needs (Table 1). In addition, we obtained information from the studies related to functions of the mobile application, measures for evaluating caregivers’ necessities, measures for evaluating the effectiveness of the mobile application and the main results (Table 2).

We also analyzed the level and degree of quality of evidence following the classification of the Scottish Intercollegiate Guidelines Network [28]. This tool was chosen due to its utility for this evaluation [29]. (Appendix A
Table A1).

To improve the quality of this study, two authors (MSG and VPJ) assessed the relevance of the studies found during the search strategy independently. They also categorized data into variables used to synthesize information from each study. Both investigators discussed discordant elements until an agreement was reached.

## 3. Results

The initial search identified 1095 references of which 214 were eliminated due to being duplicates. Following the inclusion and exclusion criteria, we identified and extracted information from 11 studies (Figure 1).

### 3.1. Study Objectives

Of the 11 studies found, eight described the design and evaluation of a mobile application for caregivers [3,9,17,20,21,23,24,27], whereas one evaluated a previously designed mobile application [22]. One of these studies subsequently developed that mobile application and published another study in which the objective was evaluated [25,26].

### 3.2. Participants

The sample sizes varied between 4 and 90 [20,21]. In some studies, the participants were only informal caregivers [3,17,20,21,22,23,24], while others used informal and formal caregivers, such as nurses and physicians [9,25,26,27].

Two studies only used female samples, such as mothers or daughters [3,20]. In the other studies, over 60% of the informal caregiver participants were women [9,22,26].

### 3.3. Situation of the Person in Need of Care

Mobile applications were designed for caregivers who take care of adults with Alzheimer’s disease or other forms of dementia [9,20,23]. Others mobile applications were designed for cancer caregivers [3,25,26,28,29], sensory processing disorder [17], premature infants [21], and cerebral palsy [22].

### 3.4. Design and Duration of the Studies

One study carried out a randomized controlled trial to evaluate if mobile apps were effective [21]. Nevertheless, most of the studies had a descriptive design [3,9,17,20,22,23,24,25,26,27].

The time that the participants tested the mobile apps varied between six months [24] and one week [27].

### 3.5. Participants Involved in the App’s Design

Of the 10 studies in which a mobile application for caregivers was designed [3,9,17,20,21,22,23,24,25,27], 8 studies considered caregivers’ needs and challenges with the aim of creating or adapting the app’s functions to them [3,9,20,22,23,24,25,27]. In addition, some studies evaluated perspectives and opinions of formal caregivers such as physicians, nurses, or nursing students [9,25,27].

### 3.6. Measures for Evaluating Caregivers’ Needs

To evaluate caregivers’ needs, the studies conducted interviews in which the family members were asked about their needs, difficulties, routines, emotional states, computer skills, and attitudes towards mHealth [3,24,25]. Furthermore, the studies carried out focus groups to ask about personal experiences in caregiving, daily challenges, expectations for the mobile application, and suggestions for content [23,25]. In addition, one of these studies interviewed formal caregivers to evaluate their opinions on possible features to include in mobile applications [25].

Some studies did not show the measures to assess care needs [9,20] or these needs were assessed in previous studies [22,27].

### 3.7. Functions of the Mobile App

The most common features of the mobile applications were the provision of information about the patient, such as sociodemographic indicators, clinical information, and activities of daily living [9,17,21,24,25,27]. Educational information and links to websites focused on caregivers [9,21,22,23,24,25,27]. Moreover, one mobile application was designed to record and send videos to other caregivers to share their experiences [20]. Contact information for caregiver resources and services was sometimes provided [9,24]. In addition, some mobile applications offered support to make decisions or to solve problems, and feedback from health professionals or caregivers could ask questions [9,17,25]. One study included a checklist with general symptoms that the caregivers should look out for to know when to call the hospital [24] and another a Behavior Problems Checklist, a validated assessment of memory-related problems, depression, and disruptive behaviors [9]. Questionnaires to evaluate caregivers’ psychological well-being [25] or track patient wellness [27] or state of mind [21], or validated questionnaires such as Zarit Burden Scale to evaluate caregiver burden, and Patient Health Questionnaire were provided to assess depression symptoms [9]. If the caregiver was experiencing high levels of depression or burden, the mobile application recommended to contact the case manager [9], and the mobile application gave information about self-care [27]. A list of medications was another common feature to help caregivers to manage treatment with reminders for taking medication [9,25]. Another feature helped to track doctors’ appointments [24,25].

Unique features of some mobile applications were meditation audio clips, communication with other caregivers [27], goal setting and tracking [17]. They sent encouragement messages as positive reinforcement [17]. Finally, a mobile application showed a tree to represent the status of the caregiver’s social network with the final aim of avoiding isolation [3].

### 3.8. Measures for Evaluating Mobile Apps

Three of the 11 studies extracted usage data of the mobile application such as the number of downloads, usage frequency, the frequency that specific features were used, the type of information searched for, and the purpose of notifications [9,17,24].

Some questionnaires were administered to evaluate the mobile application, its usefulness, satisfaction with each function and their perceived importance, ease of use, and suggestions for improvement [9]. Likewise, the mobile app rating scale evaluated the quality of app engagement, functionality, aesthetics, and the perceived impact of the app on the user’s knowledge and attitudes [27]. Other questionnaires were applied to evaluate the effects of the mobile intervention such as the Zarit Burden Scale and the Kaye’s Gain Through Group Involvement Scale to evaluate caregiver burden [20]. The App Impact Questionnaire was created to assess treatment adherence and sense of competence of parents [17]. The Parenting Sense of Competence Scale evaluated the satisfaction of parenting and parental self-efficacy [21]. The Press Ganey Discharge Questionnaire determined preparedness for discharge [21]. Moreover, one study created a questionnaire to evaluate caregivers’ knowledge [22].

Five studies carried out qualitative methods such as interviews and focus groups to evaluate the latest version of the mobile application, its usefulness, user satisfaction, helpfulness, and to offer further suggestions [3,23,24,26,27].

Finally, three studies additionally evaluated formal caregivers to assess mobile applications via focus groups, questionnaires, and interviews [9,26,27].

### 3.9. Mobile Applications Effectiveness

Of the 11 studies found, three studies evaluated the efficacy of mobile applications [17,21,22]. The results showed that treatment adherence increased significantly as a result of the use of the mobile application and the content of a mobile application [17]. In another study caregivers’ knowledge increased significantly after using a mobile application [22]. Parental self-efficacy and preparation for hospital discharge were also increased, although the differences were not statistically significant [21].

The participants claimed that the mobile application was easy to use [3,9,24,27], effective [8,17,20,21,26], useful [3,23,24,27], they would be willing to use the mobile application [3,26], and they were satisfied with it [9,23,26].

### 3.10. Best-Valued Features

The best-valued functions were the evaluation of caregivers’ psychological well-being, the possibility of contact with the case manager function, assessment of behavioral and mood disturbances function, information about available services for caregivers [9], and the use of examples and practical advice in educational videos [23]. In addition, the appearance of the mobile application was well appreciated [23].

### 3.11. Most-Used Features

The most-used features were the summary of the patient’s clinical information, in particular, the area to record blood test results, the checklist with general symptoms that the caregiver should look out for to know when to call the hospital [24], the behavior problem checklist, educational information [9], the reminders and the support to solve problems [17].

### 3.12. Recommended and Needed Features

Some suggestions about mobile applications that participants provided were: the need for a summary of patient information [24]; medication reminders and to track adherence to treatment [3,24]; appointment reminders and a shared calendar to see the doctor’s appointments [3,24]; educational information about difficult tasks [9], about medication management at home [24], about how to care for children at home, and about social and financial issues [26]; addition of pictures, videos, and other forms of multimedia in the educational information [9,26]; suggestions for activities for caregivers [3]; synchronization of other devices to share information with other caregivers [24]. Moreover, caregivers wanted to extend the use of the mobile application to friends and other family caregivers [26]

## 4. Conclusions

It is surprising how, despite the large number of studies evaluating applications designed to facilitate care, the large majority are focused on the interests of patients or formal caregivers, while informal caregiver information is the most underestimated. This review shows that most mobile applications for informal caregivers take into account users’ needs in the design of their features [3,9,20,22,23,24,25,27]. Mobile applications for caregivers are an effective solution to reduce their burden, improve their quality of life, and avoid the negative physical and psychological consequences of caring for a dependent person [1,16].

The mobile applications found were designed for caregivers of people with different diseases and situations of dependency. Most users agreed on the need to collect clinical information about the person they care for, medication reminders and doctor’s appointments reminders, as well as tracking on both, educational information about the disease and treatment, resources to help caregivers, and help with managing symptoms, contacting healthcare professionals, and sharing information with other caregivers [3,9,17,23,24,26,27].

Previous studies agreed that the main need for caregivers is getting more information [11,12,13]. For this reason, mobile applications must provide reliable information to caregivers. One of the most common practices amongst caregivers is to search for information on the internet, where sometimes the information is not secure [30].

In addition, it should be noted that just because a mobile application has more functions does not mean that it is of a higher quality; what is relevant is that it solves the complications experienced during care at home [16].

Most of the applications found, gave health support to informal caregivers related to caregiving tasks, such as remembering to give medication, offering calendars with doctor’s appointments, or giving educational information [9,17,20,21,22,23,24,25,26,27]. However, only a few of them added specific functions to cover the emotional needs of informal caregivers. Those that included them offered evaluations for their well-being, promotion of social relations and forums to share experiences [9,17,21,26,27].

Previous studies have shown that women often assume the role of caregiver. Two studies found, evaluated only female caregivers, mothers or daughters [3,29]. Even in studies where gender was not an inclusion criterion, there is a greater participation of women as informal caregivers [9,22,26].

Another previous study has reviewed the availability of mobile applications for informal caregivers on Google Play and iTunes. However, this study does not determine the efficacy or value of these mobile applications [1].

Studies have shown that mobile applications are effective [17,21,22]. Furthermore, users claim that they are easy to use and useful, and would be willing to continue using them [3,9,17,20,21,23,24,26,27]. User satisfaction and mobile applications’ effectiveness may be due to the functions that were developed based on caregivers’ needs.

However, a longer period of use of the mobile application in natural conditions is necessary to check whether the mobile application continues to be effective in the long term [31]. Moreover, experimental studies should increase the current quality of evidence and identify key elements of these interventions.

Likewise, for future studies, it is relevant to design mobile applications that coordinate patients, their caregivers, and the healthcare professionals involved to improve the quality of care that the patient receives, and increase the patients’ participation in their care [9].

Regarding the possible limitations of this study, it should be mentioned that despite having searched the most relevant databases for this subject, it is possible that other databases have not been taken into account. On the other hand, although the appropriate keywords have been used, it may be that there is a certain word from a specific area that has not been checked. Moreover, the search has been reduced to articles written in English. Another limitation to highlight is the difficulty in collecting the results due to the wide heterogeneity of the caregivers to whom the mobile applications are destined, as well as the evaluated variables of the mobile applications. Finally, as most studies are descriptive and with few participants, recommendations on these applications should be taken with caution.

To conclude, it is important to highlight the importance of developing strategies to help informal caregivers in their care tasks, as they do not have the necessary skills and knowledge. Mobile applications for caregivers can provide a solution for them. However, the effectiveness of these will depend largely on whether their characteristics match user needs.

## Figures and Tables

**Figure 1 ijerph-18-01702-f001:**
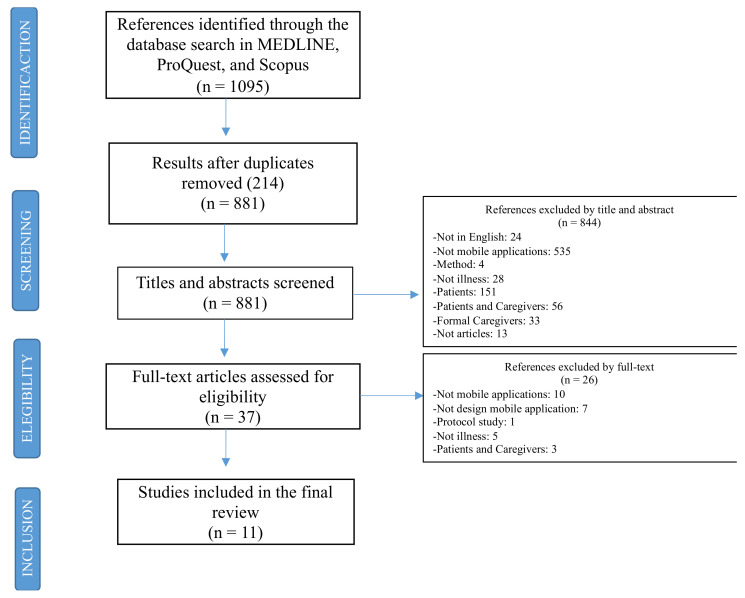
Flow diagram of study inclusion and exclusion process.

**Table 1 ijerph-18-01702-t001:** Country of the study, objective, caregivers’ participants, target groups, design, duration of the study, development of the app, and assessment of caregivers’ needs.

Authors, Year	Country	Objective	Caregivers’ Participants	Target Groups	Design—Duration of the Study	Development of the App—Assesses Caregivers’ Needs
Brown et al., 2016 [9].	USA	To describe the development and evaluate the use, perceived usefulness and potential improvements of CareHeroes, a mobile and web-based application for caregivers of people with Alzheimer’s disease (AD).	*N* = 22 11 informal caregivers (seven daughters, two sons, and two spouses), 6 home care case managers, and 5 primary care providers (PCP).	Alzheimer’s disease or other forms of dementia.	Descriptive. 11 weeks using the app.	Nurse, social worker, physical therapist, computer engineer, and health economist Yes
Davis et al., 2014 [20].	USA	To describe the development and evaluate Story-Call, a mobile application for caregivers of people with dementia.	*N* = 4 daughters caregivers Average age = 52 years	Dementia.	Descriptive. Two weeks using the app.	A linguist, a nurse, a software specialist, and a gerontologist. Yes.
Fuentes et al., 2014 [3].	México	To describe the development and evaluate EmotionMingle, a mobile application for mother caregivers of children with cancer to avoid social isolation.	*N* = 6 mother caregivers Average age = 37.2	Cancer.	Descriptive. –	Physiologists, caregivers, social workers, and patients. Yes.
Gal & Steinberg, 2018 [17].	Israel	To describe the development and evaluate SensoryTreat, a mobile application to promote adherence to home-program treatments of children with sensory processing disorders.	*N* = 45 parents of children with sensory processing disorder who were treated by occupational therapists.	Sensory processing disorder.	Descriptive. Four months using the app.	– No.
Garfield et al., 2016 [21].	USA	To describe the development and evaluate whether parents of very low birth weight infants improve greater parenting self-efficacy, their preparation for discharge and have shorter length of stay.	*N* = 90 parents. Experimental Group (E.G) = 46 Control Group (C.G) = 44	Infants with very low birth weight.	Randomized. Four weeks using the app.	– No.
Ghazisaeedi et al., 2016 [22].	Iran	To evaluate the effect of using an educational mobile application on the knowledge of the caregivers of children with cerebral palsy.	*N* = 17 caregivers of children with cerebral palsy. 82% female.	Cerebral palsy.	Descriptive. Two months using the app.	– Yes, caregivers’ needs were assessed in a previous study.
Halbach et al., 2018 [23].	Norway	To describe the development and evaluate a mobile application for caregivers of people with dementia.	*N* = 9 formal and informal caregivers.	Dementia.	Descriptive. –	– Yes.
Slater et al., 2018 [24].	Australia	To describe the development and evaluate Oncology Family, a mobile application for relatives with a child with cancer.	*N* = 38 parents and other informal caregivers.	Cancer.	Descriptive. Six months using the app.	A Pediatric Oncologist, Nurse Manager, Statewide Educator, Allied Health Clinical Leader, and Program Manager. Yes.
Wang et al., 2015 [25].	China	To describe the development of Care Assistant, a mobile application for caregivers of children with leukemia.	*N* = 23 8 informal caregivers, 12 cancer care providers, and 3 software engineers.	Leukemia.	Descriptive. –	Caregivers, oncology physicians, nurses, and engineers. Yes.
Wang et al., 2016 [26].	China	To evaluate Care Assistant, a mobile application for caregivers of children with leukemia.	*N* = 21 2 physicians, 4 nurses, and 15 informal caregivers (60% female).	Leukemia.	Descriptive. Two weeks using the app.	–
Wittenberg et al., 2019 [27].	USA	To describe the development and evaluate *Caregiver Communication about Cancer*, a mobile application for communication support informal cancer caregivers (friends or family members).	*N* = 37 11 caregivers and 26 cancer providers. In 3 study steps: 1. assessment of caregiver acceptability (*n* = 5); 2. assessment of quality and perceived impact by cancer providers (*n* = 26); and 3. acceptability testing with caregivers (*n* = 6).	Cancer.	Descriptive. One week using the app.	– Yes, caregivers’ needs were assessed in a previous study.

–Missing data.

**Table 2 ijerph-18-01702-t002:** Functions of the app, measures for evaluating caregivers’ necessities, measures for evaluating the app, and results.

Authors, Year	Functions of the App	Measures for Evaluating Caregivers’ Necessities	Measures for Evaluating the App	Results
Brown et al., 2016 [9].	-Provide patient’s information (sociodemographic indicators, clinical information, pain, activities of daily living, and instrumental activities of daily living). Send an alert to notify the case manager to call or e-mail the caregiver for non-urgent issues or questions. -Provide educational information and links to websites focused on caregivers and caregiver wellness. -Provide web links and contact information for local, state, and national caregiver resources and services. -Assess depression symptoms of caregivers (Patient Health Questionnaire-2) and burden (Zarit Burden Scale). If the caregiver’s responses indicate that he or she is experiencing high levels of depression/burden, recommendations and resources are sent to reduce those symptoms. -Provide decision support -Provide a list of patient’s medications. -Assess patient’s behavior problems (memory, depression, and disruptive behaviors). This information can be shared with professionals.	–	Collect information in real-time about the frequency that specific features were used, the types of information accessed, and the purpose of notifications. Questionnaire for informal caregivers to evaluate sociodemographic data, internet skills, average hours spent on the internet each week, the usefulness of the app, satisfaction with each function of the app, importance of each function, ease of use, suggestions for improvement and obstacles to its use. Questionnaire for primary care providers (PCP) to evaluate reasons to use the app, perception of usefulness, impact on patient outcomes and the efficiency of their practices, suggestions for improvement, and obstacles to its use. Focus group with home care providers to assess utility, experience, and suggestions for improvement.	The most used features were the Behavior Problems Checklist and educational resources. 60% of informal caregivers agreed that the app helps make decisions about new problems. More than 50% said they were satisfied with the features of the app. The most important features rated by informal caregivers were caregiver’s well-being assessment (70%), alert function for care providers (70%), the Behavior Problems Checklist (70%), and information about available services (70%). 60% of PCP anticipated CareHeroes would improve the quality of AD care (*n* = 3) and increase their professional satisfaction. The most useful features rated by case managers were educational information, caregiver self-assessment. Case managers also stated that the app was easy to use and accessible. They suggested instructional videos about difficult tasks. The challenges were lack of experience in using new technologies and the lack of time to get used to using the app.
Davis et al., 2014 [20].	Watch and record videos on care topics.	–	Zarit Burden Scale and Kaye’s Gain Through Group Involvement Scale.	Participants stated the app would allow them to manage stress and family relationships more effectively and find community health care resources.
Fuentes et al., 2014 [3].	Represent the status of an individual’s social network. -Represent the number of interactions of the caregiver with loved ones, the caregiver’s emotions, and the caregiver’s location. -Send persuasive messages to caregivers based on the interactions they have had or lost with their contacts, emotions, and lifestyle. -Connect with Facebook application to display the photos that friends upload to this network.	Interviews with a brief demographic and computer skills questionnaire, followed by questions about their routines and their emotional states.	Interviews and a focus group to evaluate the characteristics of the app.	The app prototype was positively received, and mother caregivers are open to using it. Five of the six caregivers were positive about using it themselves. They said that the visualizations in the app were appropriate and easy to understand. They said that the app helps them interact socially, but most of them did not wish to use an application to ask for help in their caregiving activities. Mothers gave recommendations such as functions to remember medication and medical appointment registrations. They would like to add or edit the intensity of the available moods.
Gal & Steinberg, 2018 [17].	-Provide a daily schedule of activities -Provide notification reminders -Provide solutions to problems -Self-monitoring -Goal setting and tracking -Feedback -Positive reinforcement (encouragement messages) -Collaboration with a therapist	–	Collect information in real-time about the usage frequency and on specific features. App Impact Questionnaire (AIQ) to assess the app’s impact on adherence and sense of competence of the parents.	The most used features were reminders, the solution to problems, and self-monitoring. A strong significant correlation betweenapp usage frequency and families’ adherence (*p* = 0.006), as well as relevancy and adherence (*p* = 0.009). Strong significant correlation between parental competence (*p* = 0.001) and usefulness of the app and relevancy of the app (*p* = 0.002).
Garfield et al., 2016 [21].	-Provide multimedia information to educate parents in childcare. -Monitoring daily living activities. -Assess daily the mood of parents.	–	Parenting Sense of Competence Scale (PSOC) PressGaney Discharge Questionnaire to assess preparedness for discharge. In addition, length-of-stay in the hospital was measured.	Parents’ self-efficacy improved (7% improvement in E.G., and <1% in C.G), although differences were not statically significant (*p* = 0.384). Preparedness for discharge was higher in E.G rather than C.G (“Very well prepared” E.G = 42%; C.G = 30%). Length-of-stay was shorter in E.G rather than C.G, although differences were not statically significant. However, parents who used the app for a longer average time had significantly shorter stay time compared to C.G (*p* = 0.085).
Ghazisaeedi et al., 2016 [22].	-Provide information about caregivers’ issues (feeding, toileting, bathing, playing, handling, and movement).	–	Questionnaire to assess sociodemographic data and caregiver knowledge about the correct daily care of children with cerebral palsy.	Caregivers’ self-assessed knowledge increased significantly after using the app in all domains (*p* < 0.05), except in the knowledge about feeding that increased, but the differences were not statistically significant.
Halbach et al., 2018 [23].	-Providing educational information in written form with videos and audios.	Focus group to document the participants’ personal experiences, the daily challenges in the relation with the person with dementia, views and expectations towards the final app, and suggestions for content and content areas.	Focus group to evaluate the final content of the app.	Participants initially outlined the areas in which they would need more information: -Medical and psychological issues -How to communicate with persons with dementia -Legal and financial issues -Practical advice for everyday challenges -Collaboration with health services Participants were satisfied with the app and found it quite useful. They stated that it is useful to distinguish between basic and in-depth information. The best value was content and appearance, regardless of the sessions’ duration, and also the use of examples and explanatory videos.
Slater et al., 2018 [24].	-Give contact information for the nearest hospital. -Report symptoms that caregivers need to consider as an indicator of when to call the hospital. -Record the results of blood tests. -Report on websites and contacts of the health care team. -Store personal notes. -Offer a calendar of doctor’s appointments.	Interviews with caregivers to confirm what features they needed in the app.	The number of app downloads. Interviews with patients, parents, and caregivers to explore app usage, the most used features, satisfaction, and suggestions to improve the app.	There were 498 downloads of the app. 68% of participants reported downloading the app. The most used features were the area for recording blood test results and the area where symptoms appear to be taken into account as an indicator of when to call the hospital. Caregivers reported the app was useful and it was easy to use. The suggestions they gave were a summary of the child’s history including date of diagnosis, date of surgery, etc., information on managing medications at home, improving the aesthetics of the app, reminders of appointments, medication reminders, and synchronizing other devices to share information between different caregivers.
Wang et al., 2015 [25] and 2016 [26].	-Provide personal information: child’s demographic data, caregiver’s age, educational level, occupation, residence, child’s diagnosis, treatment, and other clinic information. -Track treatment. -Provide solutions to problems -Provide final and social assistance. -Provide information about common symptoms that may occur during treatment. -Provide information about disease and treatment. -Provide self-assessment questionnaires to assess caregivers’ psychological well-being such as anxiety, depression, social support, care burden, and quality of life. -Ask and discuss anything with healthcare providers related to the disease and its care via Chat. -Provide reminders for medication and doctors’ appointments.	Semi-structured interviews to assess caregivers’ challenges, needs, and mHealth attitudes. Group discussion with physicians and nurses to evaluate caregivers’ challenges and needs from their perspectives, current interventions to support caregivers, perceptions of the app, and strategies to promote the caregivers’ compliance and participation. Individual interview to gather more information and suggestions. A second group discussion with caregivers to show them a provisional structure of the app. They were asked about this structure and its content, what features should be added or removed, and suggestions. A third group discussion with caregivers and engineers to transform their necessities in the app features.	Semi-structured interviews to evaluate the app’s benefits and suggestions for improvement.	Informal caregivers, physicians, and cancer nurses showed the importance of treatment follow-up. Caregivers expressed an urgent need for information and knowledge to care for their children at home, as well as information on social and financial aspects. On the other hand, they were interested in knowing their physical and psychological state. Parents were positive about their experience with the app. All caregivers expressed a willingness to continue using the app. The benefits of the app were: being more knowledgeable about the disease, more confidence in care, social support and stress reduction. Caregivers suggested adding more pictures, videos, and other forms of multimedia in the app to improve the visual and educational effect. They wanted the app to be not only for parents but also for friends and family members.
Wittenberg et al., 2019 [27].	-Talking tips about illness and self-care. -Share information and communication with the patient, family, far away family members, and healthcare providers. -Patient wellness tracker. -Meditation audio clip. -Reminders/Notes.		-Open-ended interviews to explore whether the app design was easy to use, valuable to caregivers, and likelihood of use. -Mobile app rating scale assessing ease of use, navigation, design, and layout.	Assessing acceptability of prototype: All five family caregivers found the app easy to use, the size of the print in the app accessible, and that it was “very likely” that family caregivers could follow the talking tips in the app. Provider assessment of quality and perceived impact: On the five-point scale (5 = strongly agree), providers ranked the app very likely to increase awareness of family caregiver needs (4.12), increase knowledge about communication (4.19), change attitudes toward family-centered care (4.12), increase motivation to address family caregiver concerns (4.15), and encourage family caregivers to look for help (4.15). Acceptability testing: On a 7-point scale (7 = likely), caregivers perceived the app to be useful (5.23) and had high perceived ease of use (6.00). Overall positive feedback about app acceptability.

–Missing data.

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
