# Peer review of "Mobile Apps for Helping Informal Caregivers: A Systematic Review"

_ijerph, 2021, doi:10.3390/ijerph18041702_

Round 1

Reviewer 1 Report

I thank the authors for the paper with a relevant and timely review of apps for informal care givers. Structure of the paper and design are okay, generally speaking. But I have a list of details that require your efforts for resubmission.

A. Some language and grammar issues in the following lines:

  • Some additional native English editing is required to solve all kinds of little issues.
  • For instance, be consistent in the whole text in writing past or present sense (a little example on line 161 is "consider" that should be "considered")
  • And another repeating issue is the use of "they", it is often unclear what that is refering to in the preceeding sentences (examples in lines, 166, 168, 254, 263, amongst others)
  • line 53: self-help in stead of elf-help
  • 93: word missing in "...have not a sick..." (a sick what of who?)
  • 105: This tool is single, so not tools
  • 126: Is it correct that category (D) includes both level 3, 4 AND 2+? Then combine the two now seperate sentences (but I doubt this)
  • 188: incomplete sentence
  • 232: "In other study..." should be "In another study..."
  • 236: twice "that", skipp one time
  • 244-245: unclear, incomplete sentence.
  • 257: it is "... to care for children..."
  • 270: Unclear sentence about mobile applications
  • 281: Unclear sentence about these needs 
  • 303: unclear english, "acheive for the purpose"??

B. Major issues:

  • Paragraph 2.1: why is "informal" not part of your search term "caregivers"?
  • Paragraph 2.2: was it really usefull to included Spanish in the search, as all selected papers are English, and that is common method in international journals. To be honest, I find it a bit too subjective to select a language on the basis of the background of authors. Please add a little Discussion, or skip this element. By the way: it is clear that [25] is one of the two proposal papers, which is the other one? The nature of the papers should be added to Table 1.
  • Paragraph 3, lines 128-130 and Figure 1 (major issue). It is unclear what and how you did you made your decisions on Elegibility in the step from 37 to 15 papers. And I challenge that your selected 15 papers: why proceed with the two proposals and the two designs?? You are looking for real evidence, not for designs or proposals, it is not a scoping review. The place/contribution of these particular papers in the Results section remains unclear and it is also not reflected upon in the Discussion.
  • Paragraph 3.10, lines 240-242: does all this content really stem from one single reference [8]? 
  • Paragraph 3.12, lines 252-264 should be rewritten in a good storyline. At presence it is just a bullit list of incomplete sentences.
  • Suggestion for Table 1: reframe "chronc condition" to "target groups", so that you can make explicit "older people" for papers 30 and 31 at the end of the table.
  • Table 2 is very large, perhaps in an Appendix?
  • It is remains unclear what cross-over lessons are learned in the comparison between target groups/chronic conditions.

Author Response

Dear Reviewer,

We thank the reviewer for all the comments he/she has made to improve our paper. We appreciate the time you have spent on its report, particularly in this period we assume that is an additional effort as we are aware of the difficulties arising from the COVID-19 outbreak. Many thanks. In addition, we appreciate all the suggestions and comments. We are sure they improve this work. In the manuscript, we have introduced changes in red color to make it easier to identify. In this letter, we comment one by one on these suggestions.

In addition, paper has been externally revised to improve its language and grammatical forms.

I thank the authors for the paper with a relevant and timely review of apps for informal care givers. Structure of the paper and design are okay, generally speaking. But I have a list of details that require your efforts for resubmission.

  1. Some language and grammar issues in the following lines:
  • Some additional native English editing is required to solve all kinds of little issues.

Thank you for your input. The paper has been externally revised to improve its language.

  • For instance, be consistent in the whole text in writing past or present sense (a little example on line 161 is "consider" that should be "considered")

Thanks to the reviewer for his appreciation, we have revised the verb forms of the text.

  • And another repeating issue is the use of "they", it is often unclear what that is refering to in the preceeding sentences (examples in lines, 166, 168, 254, 263, amongst others)

As suggested, the use of this term has been carefully revised and a more appropriate term has been used for each sentence.

  • line 53: self-help in stead of elf-help

Thank you for detecting the error, the word self-help has been modified.

  • 93: word missing in "...have not a sick..." (a sick what of who?)

After your suggestion, the text has been modified as follows “people who have not an illness”

  • 105: This tool is single, so not tolos

Thank you for detecting the error, the word tool has been changed.

  • 126: Is it correct that category (D) includes both level 3, 4 AND 2+? Then combine the two now seperate sentences (but I doubt this)

The text is copied literally as it appears in the Scottish guide.

  • 188: incomplete sentence

Thanks for the suggestion, the sentence has been completed as follows: Additionally, some mobile applications offered support to make decisions or to solve problems.

  • 232: "In other study..." should be "In another study..."

Thank you for detecting the error, the expression has been changed.

  • 236: twice "that", skipp one time

Thank you, the duplicated word has been removed.

  • 244-245: unclear, incomplete sentence.

After your suggestion, the text has been restructured as follows: “additionally, the appearance of the mobile application was well appreciated”

  • 257: it is "... to care for children..."

Thank you, has been added “for”

  • 270: Unclear sentence about mobile applications

As suggested, phrase has been changed

  • 281: Unclear sentence about these needs 

Thanks for the suggestion, the sentence has been changed

  • 303: unclear english, "acheive for the purpose"??

After your suggestion, the text has been restructured as follows: “Studies have shown the mobile applications are effective”

  1. Major issues:
  • Paragraph 2.1: why is "informal" not part of your search term "caregivers"?

We did not use the word "informal" in the search because most of the articles did not use it. For example, one article talks about mother caregivers, so it could have excluded some articles from the search.

  • Paragraph 2.2: was it really usefull to included Spanish in the search, as all selected papers are English, and that is common method in international journals. To be honest, I find it a bit too subjective to select a language on the basis of the background of authors. Please add a little Discussion, or skip this element. By the way: it is clear that [25] is one of the two proposal papers, which is the other one? The nature of the papers should be added to Table 1.

We have decided to skip the element of Spanish articles because there were not any of it. We found Spanish articles only with the objective of extend the search. We have decided to eliminate these references in our review because of your suggestions.

  • Paragraph 3, lines 128-130 and Figure 1 (major issue). It is unclear what and how you did you made your decisions on Elegibility in the step from 37 to 15 papers. And I challenge that your selected 15 papers: why proceed with the two proposals and the two designs?? You are looking for real evidence, not for designs or proposals, it is not a scoping review. The place/contribution of these particular papers in the Results section remains unclear and it is also not reflected upon in the Discussion.

We included a new step in Figure 1 to make it clearer. Also, according to expert comments we have decided to eliminate the two proposals and two designs in our review.

  • Suggestion for Table 1: reframe "chronic condition" to "target groups", so that you can make explicit "older people" for papers 30 and 31 at the end of the table.

We appreciate the comment and we have made changes as suggested by the reviewers

  • Table 2 is very large, perhaps in an Appendix?

We appreciate the proposal, however, we consider that the information in the table is useful to understand the study, so we submit to the editor's consideration whether it should be included in the supplementary material or appendix.

  • It is remains unclear what cross-over lessons are learned in the comparison between target groups/chronic conditions.

We changed it to make it clearer.

Reviewer 2 Report

Thank you for the opportunity to review this systematic review of mobile phone apps for helping informal caregivers in their tasks.

The title refers to “career tasks”. Are all informal caregivers doing so as a fulltime vocation i.e career? See also line 68.

A systematic review is undertaken to answer a specific question. What is the question to be answered by this review.

Methods: MeSH terms and keywords were used for the searches of Medline, ProQuest and Scopus. Of these three databases, only Medline uses MeSH terms.

Methods, 2.2: the inclusion criteria were that the paper provided results about the design of the application. Two of the papers that were reviewed, were “Proposals”, as reported in Figure 1. How can the proposal give results of the design?

Results, line 155: how does a study perform a study protocol. Presumably, one of the papers was a study protocol in which the app design was described, for it to have met the inclusion criteria?

Results lines 161-162: “11 studies consider caregivers’ necessities and challenges with the aim of adapt App’s function to them [3,8,21,23-28,31,32].” Table 2 has a column headed “Measures for evaluating caregivers’ necessities”. Thirteen papers are included in the table of which only eight have information entered in this column, and not 11. These are references, 3,23,24,26,29,30,31 and 32.

The second sentence of the Discussion states that “This review shows that most mobile applications for informal caregivers take into account future users’ needs to design their features [3,8,21,23-28,31].” (10 papers are cited). Based on the information presented in Table 2, the end-users’ needs were taken into account in only eight of 15 apps. This is just over half, and technically it is most, but the lack of appreciation of the needs of the end-users during the development of almost half of the apps is an important finding that should be discussed.

There are a number of comments and queries most of which are language related. The paper needs careful proofreading and English editing. Examples are:

In several places, the term “future caregiver users”. The word “future” appears to be redundant, as at the stage of development all users will be future users.

Line 53: should “elf-help” be “self-help”?  

Line 69: why would an app that is still to be or being developed for a specific task need to be adapted. Adapt implies that something has to be changed. See also line 162 and other places

Line 93: correct the English, “…for caregivers who have not a sick (e.g. improving young children’s nutrition).

Line 97: Excel can’t extract information from a paper. The data were extracted and entered into Excel.

Line 137 and other places: the word “other” is used when it appears that it should be “another”.

Section 3.7 contains several incomplete sentences without verbs, for example, line 188.

Line 201: “Some fewer common features” goes on to describe features listed in one paper each. They can’t be common if they are unique.

Line 223: should “focal” be “focus”.

Author Response

Dear Reviewer,

We thank the reviewer for all the comments he/she has made to improve our paper. We appreciate the time you have spent on its report, particularly in this period we assume that is an additional effort as we are aware of the difficulties arising from the COVID-19 outbreak. Many thanks. In addition, we appreciate all the suggestions and comments. We are sure they improve this work. In the manuscript, we have introduced changes in red color to make it easier to identify. In this letter, we comment one by one on these suggestions.

In addition, paper has been externally revised to improve its language and grammatical forms.

Thank you for the opportunity to review this systematic review of mobile phone apps for helping informal caregivers in their tasks.

The title refers to “career tasks”. Are all informal caregivers doing so as a fulltime vocation i.e career? See also line 68.

We appreciate the comment and change the title.

A systematic review is undertaken to answer a specific question. What is the question to be answered by this review.

The question we formulated was: What mobile applications apps are there for informal caregivers of people who are ill and do they function?, as can be seen in the objectives presented in the lines 92-94

Methods: MeSH terms and keywords were used for the searches of Medline, ProQuest and Scopus. Of these three databases, only Medline uses MeSH terms.

That is true. We have changed it by adding keywords to make it clear that not all databases use MESH. Thank you.

Methods, 2.2: the inclusion criteria were that the paper provided results about the design of the application. Two of the papers that were reviewed, were “Proposals”, as reported in Figure 1. How can the proposal give results of the design?

We appreciate the suggestion. We have decided to eliminate these studies in our review.

Results, line 155: how does a study perform a study protocol. Presumably, one of the papers was a study protocol in which the app design was described, for it to have met the inclusion criteria?

We have chosen to eliminate these studies in our review because of your suggestions.

Results lines 161-162: “11 studies consider caregivers’ necessities and challenges with the aim of adapt App’s function to them [3,8,21,23-28,31,32].” Table 2 has a column headed “Measures for evaluating caregivers’ necessities”. Thirteen papers are included in the table of which only eight have information entered in this column, and not 11. These are references, 3,23,24,26,29,30,31 and 32.

We have added the next sentence in Methods 3.6. to explain it better “There are some studies that do not show the measures to assess care needs [8,21,25,30] or these needs were assessed in previous studies [28,32]”.

The second sentence of the Discussion states that “This review shows that most mobile applications for informal caregivers take into account future users’ needs to design their features [3,8,21,23-28,31].” (10 papers are cited). Based on the information presented in Table 2, the end-users’ needs were taken into account in only eight of 15 apps. This is just over half, and technically it is most, but the lack of appreciation of the needs of the end-users during the development of almost half of the apps is an important finding that should be discussed.

Thanks for your appreciation, we have explained it better with the previous explication.

There are a number of comments and queries most of which are language related. The paper needs careful proofreading and English editing. Examples are:

In several places, the term “future caregiver users”. The word “future” appears to be redundant, as at the stage of development all users will be future users.

As suggested, the term has been reviewed and reworded where it seemed redundant.

Line 53: should “elf-help” be “self-help”?  

Thank you for detecting the error, the word self-help has been modified.

Line 69: why would an app that is still to be or being developed for a specific task need to be adapted. Adapt implies that something has to be changed. See also line 162 and other places

We have decided to eliminate these studies in our review because of your suggestions.

Line 93: correct the English, “…for caregivers who have not a sick (e.g. improving young children’s nutrition).

After your suggestion, the text has been modified as follows “people who have not an illness”.

Line 97: Excel can’t extract information from a paper. The data were extracted and entered into Excel.

We appreciate the comment and change the sentence “the information from each paper was extracted and entered into Excel program”.

Line 137 and other places: the word “other” is used when it appears that it should be “another”.

Thank you for detecting the error, the word another has been modified.

Line 201: “Some fewer common features” goes on to describe features listed in one paper each. They can’t be common if they are unique.

As suggested, the text has been modified by “unique features of each mobile application were meditation audio clips”

Line 223: should “focal” be “focus”.

Thank you for detecting the error, the word focal has been modified.

Round 2

Reviewer 1 Report

Thank you authors for carefully revising your paper and addressing the comments. I think it has become a good and relevant paper. I don't have any further comments.